# Speculative Prefill: Turbocharging TTFT with Lightweight and Training-Free Token Importance Estimation

**Jingyu Liu** [1]  **Beidi Chen** [2]  **Ce Zhang** [1]

## Abstract

Improving time-to-first-token (TTFT) is an essentially important objective in modern large language model (LLM) inference engines. Optimizing TTFT directly results in higher maximal QPS and meets the requirements of many critical applications. However, boosting TTFT is notoriously challenging since it is compute-bounded and the performance bottleneck shifts from the self-attention to the MLP part. We present SPECPREFILL[1], a training free framework that accelerates the inference TTFT for both long and medium context queries based on the following insight: LLMs are generalized enough to preserve the quality given only a *carefully chosen* subset of prompt tokens. At its core, SPECPREFILL leverages a lightweight model to speculate locally important tokens based on the context. These tokens, along with the necessary positional information, are then sent to the main model for processing. We evaluate SPECPRE-FILL with a diverse set of tasks, followed by a comprehensive benchmarking of performance improvement both in a real end-to-end setting and ablation studies. SPECPREFILL manages to serve `Llama-3.1-405B-Instruct-FP8` with up to $7\times$ maximal end-to-end QPS on real downstream tasks and $7.66\times$ TTFT improvement.

## 1. Introduction

Large Language Models (LLMs) represent a transformative innovation in artificial intelligence, enabling machines to understand and generate human-like languages (Bubeck et al., 2023; Wei et al., 2022; Feng et al., 2024). Many SOTA models have been developed, such as GPT-4 (OpenAI et al., 2024), the Llama family (Grattafiori et al., 2024), DeepSeek R1 (DeepSeek-AI et al., 2025), Mistral (Jiang et al., 2023a), Gemini (Team et al., 2024), and Qwen2 (Yang et al., 2024), to meet the increasing expectations of users. In order to broaden their real-world applications, one essential requirement is to build an efficient serving engine that can satisfy various requirements (Miao et al., 2023; Kwon et al., 2023; Zheng et al., 2024; Shoeybi et al., 2020).

There are several fundamental reasons why TTFT stands so pivotal: 1) many applications require a fast response time that directly influences how users perceive the responsiveness of the system and 2) more importantly, TTFT determines the scaling of maximal QPS an inference engine can support as shown in Figure 1. However, optimizing TTFT is an arduous task mostly because the prefill stage is largely compute-bounded and the computational bottleneck can change depending on the prompt length and batch size. For example, many works focus on improving the self-attention speed (Dao et al., 2022; Jiang et al., 2024a), but in reality, there is still a huge traffic of large-batch short to medium context queries where it is the MLP part that clogs the whole system. Despite achieving impressive results, prior works that target the prefill phase either require a post-training adaptation (Qiao et al., 2024; Horton et al., 2024) or scale less efficiently (Shi et al., 2024).

Inspired by those work, we found a key insight that LLMs can retain most of its performance when given only *a carefully chosen subset* of tokens from the prompt, and the model is able to adapt to that in a zero-shot manner. SPECPREFILL optimizes the TTFT by leveraging a secondary lightweight model to speculate locally important tokens. Only these tokens are sent later to the base model. It reduces the total FLOPS by a factor proportional to the percentage of token drop. SPECPREFILL does not require any fine-tuning and is ready to be deployed and scaled to larger models. We summarize our key contributions:

- We present a conceptually simple, effective, and noval framework called SPECPREFILL that significantly

---

[1]Department of Computer Science, The University of Chicago, Chicago, IL, USA [2]Department of Electrical and Computer Engineering, Carnegie Mellon University, Pittsburgh, PA, USA. Correspondence to: Jingyu Liu <jingyu6@uchicago.edu>, Ce Zhang <cez@uchicago.edu>.

*Proceedings of the $42^{nd}$ International Conference on Machine Learning*, Vancouver, Canada. PMLR 267, 2025. Copyright 2025 by the author(s).

[1]The code with experiment reproduction is available at *https://github.com/anonymous/speculative_prefill*.

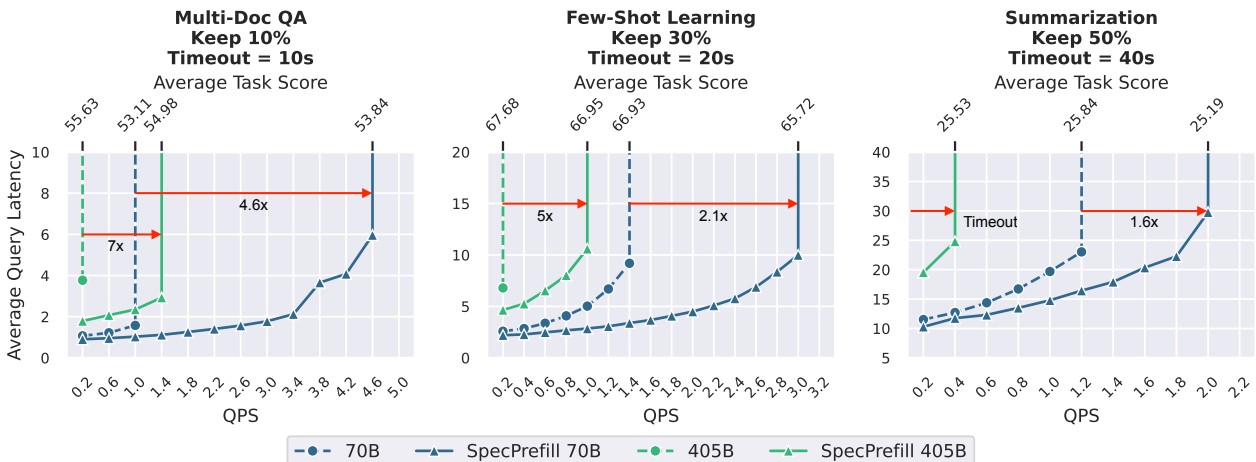

Figure 1. **Speculative Prefill QPS Improvement:** In an end-to-end server-client setting with real world datasets, we benchmark the average query latency under a given fixed timeout when sending queries at a constant QPS. SPECPREFILL significantly improves the maximum QPS supported by the vLLM server as well as the latency compared to not using it. When we reach low keep rate, we can even serve the 405B model with SPECPREFILL to run more efficiently than the 70B model. As the base model size increases and keep rate drops, we can get $7\times$ end-to-end QPS boost while only occurring $< 5\%$ accuracy.

improve the prefill phase, hence the maximal QPS, of LLM inference without any fine-tuning or adaptation.

- We conducted comprehensive evaluations both on real and synthetic datasets to demonstrate its effectiveness and limitations, giving a full picture of the expected benefits when deployed to productions.

- We implemented our method on industry standard serving engines, and benchmark its performance in both an end-to-end fashion and ablation experiments. The end result is a system that can serve `Llama-3.1-405B-Instruct-FP8` with up to $7\times$ maximal QPS under the same system specification and $7.66\times$ reduced TTFT while maintaining decent accuracy.

- Our method can be easily combined with techniques from quantization, KV eviction, and speculative decoding, making it an ideal add-on to existing engines.

## 2. Background

### 2.1. Inference Bottlenecks

Improving LLM inference efficiency has been extensively studied in prior work (Miao et al., 2023; Yuan et al., 2024). We review works that focus on different aspects when dealing with real serving systems where the bottlenecks are quite different under various serving requirements (e.g. long context domains, latency sensitive applications, etc) (Kwon et al., 2023; Zheng et al., 2024).

LLM inference can be roughly divided into two major procedures, namely the prefill phase where the model computes

the KV necessary for producing the output based on the query and the decoding phase where the model predicts new token auto-regressively.

### 2.2. Decoding Acceleration

The decoding phase is mostly memory-bounded, and therefore reducing the amount of data to move around will effectively help improve the latency. As a result, explicitly manipulating the KV cache has been extremely successful with many strategies: H2O (Zhang et al., 2023) and StreamingLLM (Xiao et al., 2024b) idenfitied key insights to KV dynamics which is used to evict less essential KV caches during decoding. CacheGen (Liu et al., 2024), Q-Hitter (Zhang et al., 2024b), and ShadowKV (Sun et al., 2024) apply efficient techniques to compress/quantize, store, and transmit KV caches to reduce memory overhead. Speculative decoding relies on the insight that hides the memory latency by concurrently verifying several speculated tokens from either a draft model or itself (Leviathan et al., 2023; Zhang et al., 2024a; Xia et al., 2024).

Despite being crucially important, decoding speed is not the only factor that influences the overall inference pipeline and we will review why *sometimes the prefill time optimization is even more essential* in many cases.

### 2.3. Prefill Acceleration

The time-to-first-token (TTFT) is crucially important both from a user experience but also the system serving perspective (Sec 4.7). In many critical applications, the input token length can often eclipse that of the generation tokens (e.g. 10:1 ratio) in real traffic (Qiao et al., 2024). Unlike

the decoding phase, however, the prefill phase is usually compute-bounded and the cost for the MLP calculation and the communication of tensor parallelism quickly becomes a bottleneck.

Many prior works have explored ways to make self-attention faster: Flash-attention series (Dao et al., 2022; Dao, 2023) compute the exact attention using carefully designed hardware-aware algorithms. Special (both static and dynamic) attention masks are designed for sparse calculation such as LongFormer (Beltagy et al., 2020), MInference (Jiang et al., 2024a), FlexPrefill (Lai et al., 2025), Hip Attention (Lee et al., 2025), Sample Attention (Zhu et al., 2024), and Duo Attention (Xiao et al., 2024a). However, none of these directly make the MLP part faster like SPECPREFILL. SPECPREFILL achieves consistent efficiency improvements in various regimes because it skips parts of the attention + MLP calculation and the all-reduce overhead, which proves to be effective especially when the ratio of $\frac{\text{batch size}}{\text{sequence length}}$ is large (Xiong et al., 2023).

Orthogonal to techniques such as prompt compression/rewrite (Jiang et al., 2023b; 2024b; Li et al., 2023), layer dropping (Elhoushi et al., 2024), and weight quantization methods (Lin et al., 2024), we explore selecting *important* prompt tokens to skip the full forward computation. GemFilter (Shi et al., 2024) uses an extra pass to get a model's own middle layer attention information that decides on what tokens to keep for the real forward. Contrast to this, we apply a separate and cheaper model to speculate locally important tokens via token transferability, which can scale more efficiently than GemFilter. Concurrent to ours, SwiftKV (Qiao et al., 2024) learns to skip later layers by reusing the past layers' KV, which achieves up to $50\%$ TTFT reduction (SPECPREFILL can reach up to $87\%$ TTFT reduction). Unlike our zero-shot requirement, they require extra light-weight fine-tuning due to modified model behavior. It is worth noting that our method approaches the problem in a different way, which makes them complimentary to each other. Finally, akin to our motivation, KV Prediction (Horton et al., 2024) proposes to adapt a cheaper system (i.e. a learned auxiliary network and a KV predictor) to predict the KV cache of the base model, thus bypassing the original KV computation. We show that SPECPREFILL can accomplish better TTFT reduction than theirs, without introducing extra overhead when coupled with speculative decoding (Leviathan et al., 2023) while maintaining competitive quality.

## 3. Speculative Prefill

In this section, we present SPECPREFILL by first describing its high-level algorithm, followed by several design choices that mitigate various biases, and a detailed implementation account. Finally, we touch a bit on how to integrate SPECPREFILL to speculative decoding, forming a full small-model-assisted inference paradigm.

### 3.1. Overall Architecture

SPECPREFILL follows a conceptually simple architecture where a usually less expensive model is chosen as the speculator model that predicts contextually important tokens given a prompt. The speculated tokens, alone with the original position information, are then fed to the main model for processing. In the following section, we will discuss two central design choices in more details, namely the token estimation algorithm and the selection strategy. Note that SPECPREFILL can be seamlessly integrated with speculative decoding in which the small model can work both in the prefill stage for token selection and the decoding stage for drafting proposals, making our approach almost free to integrate and deploy.

### 3.2. Token Importance Speculation

The goal here is to select which tokens are contextually important for a given query and send those along with necessary positional information for the main model. The procedure starts with calculating the attention scores from the speculator, which uses the last token's attention score w.r.t. the context as the surrogate for measuring token importance:

$$a_{ij} := \text{Softmax}(Q_{M+j}K^T))_i, \forall 0 \le i < M, 0 \le j < N$$

where $M$ is the context length, $N$ is the number of look-ahead steps, and $a_{ij}$ is the attention score for the $i$th token in the prompt w.r.t. the $j$th decoded token, assuming we're looking at a particular layer.

We build on top of this by aggregating the scores over the whole speculator model (Section 3.2.2) with potential look-ahead (Section 3.2.1) and select tokens based on chunks (Section 3.2.3). The subset of chosen tokens with their original positional information (Section 3.2.4) will then be used for the main model's inference.

#### 3.2.1. MITIGATE POSITION BIAS VIA LOOK-AHEAD

Prior works have shown that there are many biases for attention scores, such as the sink phenomenon (Xiao et al., 2024b) (the first couple of tokens tend to have higher weights) and the proximity bias (tokens closer to the output tend to have higher weights (Lv et al., 2024)). To mitigate these issues, instead of relying on the attention score of the last token alone, we further decode the speculator by $N$ steps and obtain the attention information from the new $N$ tokens (Wan et al., 2024). $N$ here serves as a trade-off between bias and budget, which can substantially increase the performance for shorter context queries.

### 3.2.2. Aggregated Attention Score as Token Importance

Given the full attention scores of the speculator, we decide to use a max-mean aggregation strategy to map to scalar token importance. Formally, given an attention score tensor of shape $[N, L, S, H]$ where $N$ is the number of look-ahead tokens, $L$ is the number of layers, $S$ is the sequence length, and $H$ is the number of heads, we take the maximum over $H$ and $L$ dimension to make salient tokens stand out, and average over $N$ to account for fair token contribution.

### 3.2.3. Denoise Attention Scores by Chunk Selection and Pooling

It has also been observed in concurrent works (Lv et al., 2024) that tokens that are positioned nearby share similarity in importance. We take this insights to select tokens by chunks in order to reduce the variance of our token importance estimation. Specifically, we chunk the context contiguously and average the token score within each block, and then we select the Top-K blocks. In order to eliminate the artifacts of chunkation, we apply a 1D average pooling before this to smooth the cross block scores.

### 3.2.4. Restoration of Position IDs

Finally, when we select the subset of tokens based on our compute budget and query compressibility, we also need to restore the position information which are also sent to the main model. Basically, instead of using a contiguous position ids as before, we send a potentially non-continuously increasing position ids which are obtained from tokens' positions in the original context. In addition to that, we also need to explicitly set the decoding token position to the context length in case we dropped tokens before the first decoding token. An example is shown below with ten prompt tokens and three decoding tokens (bold):

$$\text{Original Pos Ids: } [0, 1, 2, 3, 4, 5, 6, 7, 8, 9]$$
$$\text{Speculated Pos Ids: } [0, 1, 3, 6, 7]$$
$$\text{Decoding Pos Ids: } [0, 1, 3, 6, 7, \mathbf{10, 11, 12, ...}]$$

where the **bold indices** are the decoding positions which are offset based on the original position information. We found this design choice to be crucially essential, especially for position-sensitive tasks such as synthetic tasks involving retrieval and counting.

### 3.3. Implementation Details

We describe both the high-level procedure and the implementation details of SPECPREFILL in this section. In Algorithm 1, we list the high-level steps of conducting SPECPREFILL. Our implementation is based on creating a monkey patch on top of vLLM (Kwon et al., 2023) which only needs a few line of code along with a configuration file to enable SPECPREFILL. The KV cache is not necessary if we do not need to look-ahead for our speculator, which can save lots of memory allocation. However, we do need to explicitly store the queries of the decoded tokens (including the last token of the input query) which we later retrieve to compute the attention score. Note that a specific mapping (e.g. slot mapping in vLLM) might be kept track of to retrieve the right data. For batched look-ahead, we only consider tokens that are valid by checking afterwards whether they are equal to the EOS tokens. Finally, we want to mention that despite being being a sequential implementation, we can actually split the process of speculation into a separate procedure and decouple from the inference of the main model by adding a new layer of scheduling, which we leave as a future work.

### 3.4. Relation to Speculative Decoding

Speculative decoding has been proven to be extremely successful at accelerating the decoding TPS. SPECPREFILL can be seamlessly combined with speculative decoding by sharing the same draft model. Since speculative decoding itself requires a full forward pass of the context, SPECPREFILL will provide the necessary KV information required for subsequent decoding speculation, hence amortizing the overhead. This will open-up a huge space of possibilities, and lead to the first paradigm of an inference system that is fully aided by smaller speculators.

---

**Algorithm 1** Speculative Prefill

---

**Require:** Base model $M$, speculator $S$, look-ahead steps $N$, batch of mixed requests $B$, base model QKV cache $C_b$, speculator KV cache $C_s$
1: $B_p, B_d \leftarrow split\_prefill\_decode\_requests\ (B)$
2: {Section 3.2.1}
3: **for** $i = 1$ to $N$ **do**
4: $\quad B_p' \leftarrow model\_forward\ (S, B_p, C_s, store\_q\text{=True})$
5: $\quad B_p \leftarrow update\_requests\ (B_p, B_p')$
6: $\quad B_p \leftarrow check\_for\_eos\ (B_p)$
7: **end for**
8: **if** $is\_tensor\_paralleled\ ()$ **then**
9: $\quad tp\_gather\_qk\ (C_s)$
10: **end if**
11: $Q, K \leftarrow retrieve\_qk\ (B_p, C_s)$
12: $A \leftarrow compute\_attention\_score\ (Q, K)$
13: {Section 3.2.2}
14: $A \leftarrow aggregate\_attention\_score\ (A)$
15: {Section 3.2.3}
16: $T \leftarrow chunk\_select\_from\_smoothed\_attention\ (A)$
17: {Section 3.2.4}
18: $P \leftarrow restore\_pos\_ids\ (T, B_p)$
19: $B \leftarrow merge\_requests\ (T, P, B_p, B_d)$
20: **Return** $model\_forward\ (M, B, C_b)$

---

# 4. Experiments

In this section, we start with our experiment setup for reproducibility, followed by categorizing prompt compressibility of different queries. We evaluate SPECPREFILL on downstream long context, synthetic context probing, and standard short tasks. Finally, we conclude with a comprehensive efficiency measurement of our system under the real end-to-end setting.

## 4.1. Setup

We implement SPECPREFILL in vLLM that supports tensor parallelism with the same degree as the main model[2]. Due to its token dropping nature, we focus on evaluating *generative* tasks in this section and include a comprehensive range of benchmarks to fully present its applicability and potential pitfalls. We run all of experiments using a tensor parallelism of 8 for both the speculator and the base model across either 8 NVIDIA H100s or H200s (full system specification in Appendix D and guidance on reproducing results in Appendix C). We choose LLAMA-3.1-8B-INSTRUCT (Grattafiori et al., 2024) with BF16 precision as our speculator for a balance of efficiency and attention transferability and couple it with either `Llama-3.1-70B-Instruct` (BF16) or `Llama-3.1-405B-Instruct-FP8`[3] (fully quantized FP8) as the base model. In terms of token keep rate, we use a fixed percentage (i.e. the ratio of chunks when we do chunk selection) for a given task. In practice, we might devise more adaptive strategy for how many tokens to keep based on the query compressibility discussed next, or delegate the decision to users based on their needs. We leave all these possibilities for prospective applications.

## 4.2. Query Context Compressibility

We empirically found three types of queries during our evaluations based on the quality difference before and after applying SPECPREFILL:

1. *Information-dense queries*: These queries usually are short and information dense, which naturally makes token dropping less effective because there is no redundancy in the prompt.

2. *Compressible queries*: These queries are those that do not get degradation after removing a significant amount of tokens, often seen in long context tasks.

3. *Noisy queries*: These queries, perhaps surprisingly, get better results after dropping some "noisy" tokens.

We hypothesize the reason behind the improvement might be that SPECPREFILL helps remove noisy and distracting tokens in the prompt, hence projecting the prompt to the space where the main model performs better.

We will see examples in each categories in the following evaluations. It is worth noting that we used a fixed keep percentage for our evaluation and it can be tremendously helpful to automatically decide on the percentage based on the query, pushing the limit of SPECPREFILL, which we leave as a future work.

## 4.3. Baselines and SPECPREFILL Variants

We aim to showcase both the quality and efficiency of SPECPREFILL under a comprehensive set of applications. To do so, we compare *three* variants of SPECPREFILL against *four* baselines:

- *Baselines*: We compare SPECPREFILL against four different baselines: 1) **Base Llama instruct model**. 2) **Sentence RAG**: SPECPREFILL can be framed as a special case of retrieval-augmented (RAG) LLM with the granularity of tokens or blocks and the relevance metric controlled by the speculator's internal knowledge (Li et al., 2024; Gao et al., 2024b; Lewis et al., 2021). Therefore, we implemented two simple sentence-level RAG baselines and report the better one as RAG-LLAMA. 3) **LLMLingua** (Jiang et al., 2023b): SPECPREFILL can also be seen as a context compression technique, and hence we test SPECPREFILL against a text-level compression method. 4) **MInference** (Jiang et al., 2024a): To understand the benefits of skipping the MLP part, we include a sparse attention optimization approach for completeness.

- SPECPREFILL: SPECPREFILL with raw attention scores and ignoring the techniques we discussed in Section 3.2.1 and 3.2.3.

- SPECPREFILL *Full*: SPECPREFILL with all techniques but no look-ahead.

- SPECPREFILL *Full LAH*: SPECPREFILL with all techniques with 8-step look-ahead[4].

## 4.4. Real Long Context Tasks: LongBench

We start with long context tasks using LongBench (Bai et al., 2024), which consists of six different categories focusing on various aspects of long context modeling abilities.

---

[2]We expose the API so that it only takes a few line of code to apply SPECPREFILL before initializing vLLM engines.

[3]https://huggingface.co/neuralmagic/Meta-Llama-3.1-405B-Instruct-FP8.

[4]We empirically found that going beyond 16 look-ahead gives minimal performance gain.

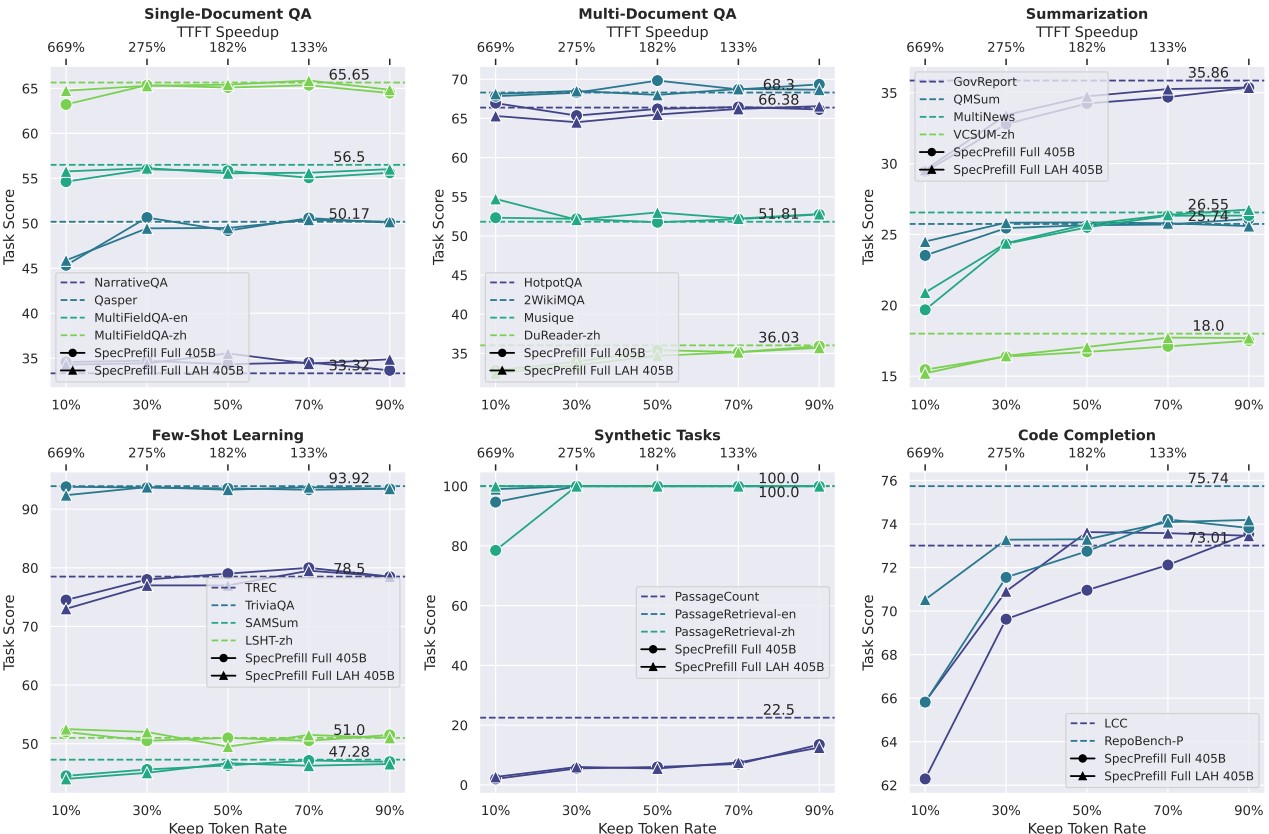

*Figure 2.* **LongBench Main Result on Llama 405B:** In this figure, we showcase the effectiveness of SPECPREFILL on Long-Bench, which consists of six categories of long context downstream tasks. In each plot, the dash lines are the results of baseline `Llama-3.1-405B-Instruct-FP8` for each subtask and we benchmark SPECPREFILL with increasing token keep rates. We observe different behaviors such as quality preservation, degradation, and improvement based on the task type.

In Figure 2, we report the main results on LongBench for `Llama-3.1-405B-Instruct-FP8`, whose length information is visualized in Appendix Figure 10. To compliment it, we also include the performance of `Llama-3.1-70B-Instruct` in Appendix Figure 8. We vary the token keep percentage starting from 10% to 90% and draw the baseline model quality using the dash lines for each subtask. To ablate the effect of our design choices, we compare SPECPREFILL and SPECPREFILL *Full LAH* with the 70B models in Appendix Figure 8, and SPECPREFILL *Full* and SPECPREFILL *Full LAH* with the 405B model in Figure 2.

As we can observe, for categories such as Single-Document QA, Multi-Document QA, Few-Shot Learning, SPECPRE-FILL can preserve most of the quality up to keeping only 10% tokens. For Summarization, we expect to see some degradation in performance as we drop more. Perhaps surprisingly, for the smaller 70B model, we can achieve better quality after we remove some tokens on tasks like Code Completion. As the model size increases, the quality gap

between applying SPECPREFILL or not becomes smaller, which indicates that bigger models adapt better with our speculated subset of tokens.

To ablate the effectiveness of techniques discussed in Section 3.2, we compare them separately in Figure 2 and 8 to avoid crowdedness. In both cases, we can see consistent improvement and the benefits of look-ahead are more consistent in shorter context tasks (more details in Sec 4.6).

Finally, we demonstrate the superiority of SPECPREFILL over three different baselines in terms of preserving the quality of the inference for the 70B model. In Table 1, we group tasks in LongBench into categories and list the results with varying degrees of compression rates. For RAG-LLAMA, we use the question to retrieve the relative information from the context (more detailed descriptions are given in Appendix B). For LLMLingua, we follow their official examples and only compress the context, leaving the question and template intact. With the prior knowledge of separated context and question, these two methods are eclipsed by

*Table 1.* **LongBench 70B Model Comparison:** We compare different methods based on `Llama-70B-Inst` with varying compression rates on LongBench (grouped by task types). * denotes models that not only require context-question separation but also have the true compression rates distinctive from the predefined ones. Among all tested methods, SPECPREFILL achieves superior average performance (underlined scores are the best under the comparable rate).

| Model | Compression Rate | Single-Doc QA | Multi-Doc QA | Sum | Few-shot Learning | Code | Synthetic | Avg |
|---|---|---|---|---|---|---|---|---|
| Baseline | N/A | 50.57 | 53.11 | 25.84 | 66.93 | 52.33 | 72.50 | 53.55 |
| RAG* | 10.38% | 32.32 | 41.17 | 18.86 | 45.40 | 44.76 | 30.42 | 35.49 |
| | 27.68% | 38.43 | 47.41 | 21.42 | 50.53 | 45.80 | 35.50 | 39.85 |
| | 45.64% | 40.53 | 46.64 | 22.45 | 49.52 | 46.00 | 43.15 | 41.38 |
| | 63.42% | 41.40 | 47.43 | 23.30 | 52.21 | 46.19 | 47.22 | 42.96 |
| | 82.22% | 43.25 | 48.16 | 23.56 | 51.44 | 45.92 | 53.04 | 44.23 |
| LLMLingua* | ~10% | 26.50 | 32.94 | 20.95 | 37.40 | 45.00 | 16.33 | 29.85 |
| | ~30% | 38.83 | 44.02 | 23.37 | 42.23 | 47.27 | 37.00 | 38.79 |
| | ~50% | 43.64 | 50.67 | 24.77 | 50.96 | 49.05 | 60.33 | 46.57 |
| | ~70% | 45.90 | 52.88 | 25.44 | 59.77 | 51.48 | 68.50 | 50.66 |
| | ~90% | 45.94 | 53.91 | 25.87 | 60.46 | 54.06 | 72.00 | 52.04 |
| MInference | N/A (Section 4.7.2) | 50.46 | 53.23 | 25.83 | 66.36 | 52.48 | 69.00 | 52.89 |
| SPECPREFILL | 10% | 47.64 | 52.96 | 21.74 | 64.52 | 63.33 | 66.25 | 52.74 |
| | 30% | 49.47 | 53.39 | 24.41 | 65.83 | 62.62 | 67.83 | 53.92 |
| | 50% | 50.18 | 52.56 | 25.10 | 65.60 | 59.91 | 68.17 | 53.59 |
| | 70% | 50.06 | 52.44 | 25.51 | 65.77 | 58.08 | 68.67 | 53.42 |
| | 90% | 50.26 | 53.25 | 25.65 | 66.35 | 53.47 | 70.67 | 53.27 |

SPECPREFILL by a large margin under the same rate. For MInference, we use the official searched optimal pattern, and SPECPREFILL can reach 99.7% average score with only 10% keep rate and outperform it with larger keep rate. Since the exact token-level "keep rate" of a sparse attention kernel is not defined, we defer to Section 4.7.2 for a more fair comparison between these two approaches. Overall, SPECPREFILL achieves impressive performance without any fine-tuning or input assumption, further supporting its effectiveness and flexibility.

### 4.5. Synthetic Context Probing: RULER

In addition to LongBench, we also evaluate SPECPREFILL on a synthetic context probing task to see if SPECPREFILL can preserve effective context lengths. RULER (Hsieh et al., 2024) is a suite of synthetically created tasks with controllable lengths, which ranges from retrieval, multi-hop tracking, real QA datasets, and context aggregation tasks. In Table 2, we include the results for the 70B model with SPECPREFILL that keeps 10% context. As we can observe, SPECPREFILL preserves the quality despite only using one tenth of the tokens except for aggregation tasks, which we believe to fall into the category of information-dense queries that are not our main target application. Take CWE from aggregation category for example: CWE asks for the common words presented in the prompt, which becomes challenging to answer by token dropping. We hope to explore in the future ways of potentially rewriting the queries instead of directly dropping the tokens to mitigate this type of limitation (Jiang et al., 2023b; 2024b). Averaging scores without the aggregation task, we can see that in most context lengths,

SPECPREFILL even helps improve the quality[5], suggesting 1) that SPECPREFILL provides both efficiency and performance gains at the same time, and 2) the fact that there are lots of potential redundancy and noise in these synthetic tasks.

### 4.6. Standard Short Tasks

Unlike prior works on prefill token dropping techniques (Lv et al., 2024; Shi et al., 2024) that do not include regular short context task evaluation, we present a wide range of standard tasks to show the full spectrum of SPECPREFILL's performance and potential caveats. We select tasks spanning general knowledge (Generative MMLU (Hendrycks et al., 2021) and Instruction Following Evaluation (Zhou et al., 2023)), math (GSM8K 8 Shots (Cobbe et al., 2021)), coding (HumanEval (Chen et al., 2021) and MBPP (Austin et al., 2021)), and reasoning abilities (Arc Challenge (Clark et al., 2018) and GPQA 8 Shots (Rein et al., 2023)).

In Apendix Figure 6, we showcase the performance of `Llama-3.1-70B-Instruct` on these tasks. Non-surprisingly, prompts from standard tasks without few shot examples are very information dense, making SPECPREFILL less effective with low token keep rate. However, for certain tasks (e.g. MBPP and GPQA), we do observe improved performance when dropping certain tokens. On average, SPECPREFILL can maintain and even surpass the baseline when choosing the right token keep rate.

---

[5]For 4k Multi-hop Tracking, since we only keep around 400 tokens, we might unintentionally ignore some essential information. But to keep experiment setup more consistent, we list the results here for clarity.

*Table 2.* **RULER Results on Llama 70B:** We present results of SPECPREFILL with 10% token keep rate on the effective context probing suite RULER with varying context length. SPECPREFILL can preserve the performance of all except for aggregation tasks, which are *less compressible* due to the problem nature as each word in the prompt is important to reason about word frequency and commonality.

| Model Name | Task Length | Retrieval Niah Variants | Multi-hop Tracking Variable Checking | QA SQuAD & HotpotQA | Aggregation CWE & FWE | Average w/o Aggregation |
|---|---|---|---|---|---|---|
| Llama-70B-Inst | 4k | 100.0 | 100.0 | 76.9 | 99.7 | 92.3 |
| | 8K | 99.9 | 100.0 | 74.7 | 98.0 | 91.5 |
| | 16K | 99.8 | 100.0 | 72.0 | 97.8 | 90.6 |
| | 32K | 99.6 | 100.0 | 69.8 | 96.9 | 89.8 |
| | 64K | 98.5 | 99.9 | 65.1 | 65.6 | 87.9 |
| | 128K | 76.5 | 56.1 | 48.2 | 41.3 | 60.3 |
| SPECPREFILL with 10% Keep Rate | 4K | 99.7 | 89.6 | 75.2 | 77.9 | 88.2 |
| | 8K | 99.6 | 100.0 | 75.6 | 79.7 | 91.7 |
| | 16K | 99.5 | 99.1 | 75.3 | 78.5 | 91.3 |
| | 32K | 99.7 | 100.0 | 72.6 | 70.0 | 90.8 |
| | 64K | 99.5 | 99.8 | 71.9 | 54.9 | 90.4 |
| | 128K | 85.8 | 55.6 | 55.3 | 48.3 | 65.6 |

## 4.7. Efficiency Benchmarking

SPECPREFILL offers great improvement to TTFT, a speedup almost proportional to the percentage of tokens we drop from the speculator, with almost ignorable overhead as we increase the base model size. In this section, we benchmark both the 70B and 405B models under two settings: 1) understanding the average query latency and QPS dynamics with real downstream datasets, and 2) evaluating TTFT with varying sequence lengths on synthetic data. We used one node consisting of eight NVIDIA H200s for all experiments unless separately specified (full system specification is listed in Table 4 from the Appendix D).

### 4.7.1. AVERAGE QUERY LATENCY UNDER DIFFERENT QPS WITH REAL DOWNSTREAM DATASETS

We want to measure the *real* performance gain we can create in an end-to-end fashion. To do this, we launch a vLLM server with a given model, and an OpenAI API client [6] that sends asynchronous requests at a constant QPS with queries from datasets in LongBench. We measure the client-side per query latency that consists of the prefill stage with several decoding steps based on the maximum budget defined by the task. In Figure 1, we increase the QPS of our client and calculate the average query latency with a given fixed timeout to simulate real-world user needs. For each task category, we draw samples randomly from each subtask and shuffle them before starting the querying, making sure the same set of queries is used over all QPS. As we can observe, all models will follow a standard three-stage pattern: 1) the initial constant stage where latency remains almost unchanged as all queries can be finished before receiving new ones, 2) the middle linear stage where TTFT is small enough but the decoding step might not finish fast enough, and 3) the final timeout stage where the server can not even finish the prefill stage before new requests and all subsequent queries are jammed thereafter. Since the maximum QPS

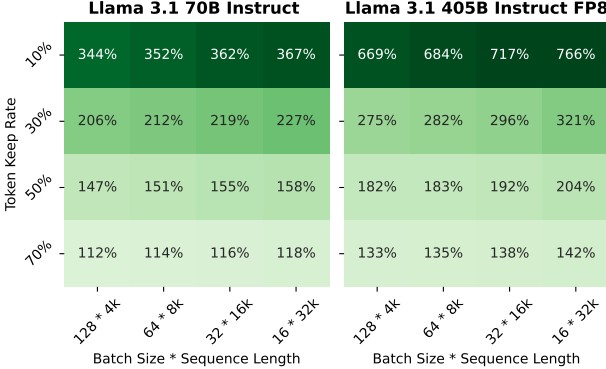

*Figure 3.* **SPECPREFILL TTFT Improvement:** We present prefill TTFT speed-up using SPECPREFILL under different settings over Llama-3.1-70B-Instruct and Llama-3.1-405B-Instruct-FP8 (achieving up to 7.66x faster TTFT when keeping 10% tokens for the 405B model).

a system can support is $\mathcal{O}(1/TTFT)$ [7] given finite timeout, the acceleration from SPECPREFILL will drastically increases the maximal QPS under a fixed timeout, which pushes the transition from stage 2 to stage 3 further later.

We show that with the help of SPECPREFILL, the 405B model can convert to $7\times$ QPS improvement on a Multi-Doc QA suite from LongBench while maintaining $> 95\%$ accuracy. The results vary as we change the model FLOPS ratio and drop rate. We believe that this type of analysis provides invaluable insights and tangible benefits of SPECPREFILL when deployed to real world systems.

### 4.7.2. TTFT IMPROVEMENT OVER DIFFERENT BATCH SIZE × SEQUENCE LENGTH PRODUCTS

We try to understand the dynamics of SPECPREFILL under different batch-size-sequence-length products and keep per-

---

[6] https://github.com/openai/openai-python

[7] If we have finitely large timeout, we would expect SPECPREFILL to support around $N$ times larger maximal QPS if we reduce TTFT by $N$ times.

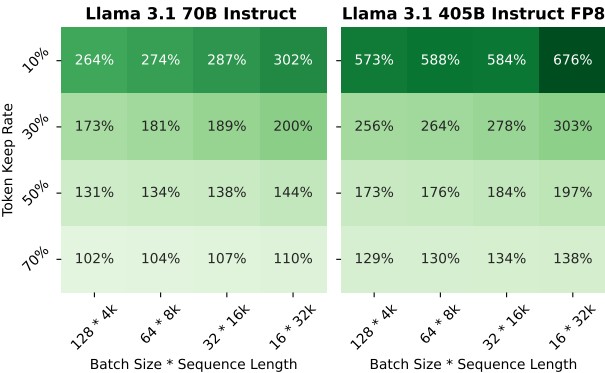

Figure 4. **SPECPREFILL with look-ahead TTFT Improvement:** Complimentary to Figure 3, we also show the relative speedup when using a look-ahead = 8 steps for both the 70B and 405B model.

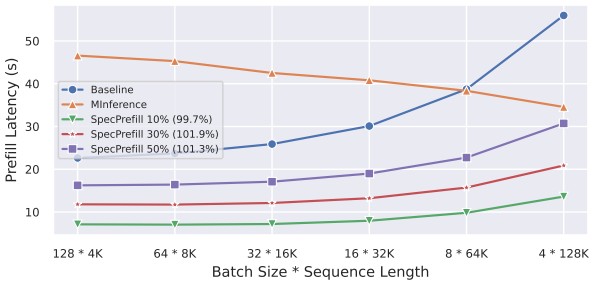

Figure 5. **SPECPREFILL v.s. MInference TTFT on 70B Models:** The superiority of SPECPREFILL becomes more clear as we increase the batch size under 128K context lengths and MInference gradually improves as the context length increases with smaller batch size due to less overhead. Percentages in parenthesis are the relative average scores to that of MInference on LongBench.

centage while isolating the advantage of TTFT. We use the official script from vLLM for latency benchmarking. In Figure 3 and 4.7.2 (with look-ahead), we highlight the TTFT speedup against the vanilla base model without SPECPREFILL, which we produce by setting the maximum decoding step to be 1. As we can see, for both the 70B and 405B models, not only do we see more direct effects of SPECPREFILL but also imply the increasing scaling along the sequence dimension. As the relative FLOPS ratio between the speculator and main model becomes larger, the overhead of speculation starts to become more negligible, which leads to more substantial improvement. In order to better understand the whole system, we include theoretical analysis of the overhead and the performance gains in Appendix E.

One salient feature of SPECPREFILL is the ability to skip some MLP calculations, one of the major benefits of which is its strong performance under large batch size. To further understand the trade-offs, we compare the TTFT of SPECPREFILL and MInference using the 70B model (one

node of 8xH100s with TP=8). In Figure 4.7.2, we can see that SPECPREFILL outperforms both the dense model and MInference when using large batch size and short to medium length prompts (i.e. less than 128K tokens). The main limitation for MInference under this scenario is the overhead introduced due to additional approximations, which gets amortized only when the ratio $\frac{\text{sequence length}}{\text{batch size}}$ becomes large enough (Jiang et al., 2024a). Overall, we found SPECPREFILL to be able to achieve $2.54\times$ to $6.54\times$ relative speedup over MInference with $99.5\%$ quality performance. We believe that this experiment provides a more comprehensive guide to practitioners on which method to choose for specific application needs (i.e. large batches plus less than 128K or ultra long prompts).

## 5. Limitation

The main limitation of SPECPREFILL is akin to all token-dropping based method: 1) they do not support explicit logit outputs for all input tokens, and hence we focus on generative evaluation, 2) without explicit context recomputation, multi-turn conversation can potentially fail (Li et al., 2025), which poses a trade-off of whether we should compute and store all KV caches during the prefill. Given that we have the full knowledge of the speculator, we advertise for more principled method to estimate token importance beyond attention scores and efficient methods for dynamic KV recomputation. Finally, we believe that a robust algorithm that determines how many tokens are required for a given prompt will be of great use for SPECPREFILL.

## 6. Conclusion

In this work, we introduce SPECPREFILL, a training-free framework for accelerating the LLM inference by speculating what tokens to drop with the help of a smaller speculator model. Leveraging the insight that models of different sizes within the same family can usually transfer token importance, SPECPREFILL not only achieves substantial improvement on TTFT, which leads to $7\times$ maximal supported QPS of an inference system, but also reduces the memory required. SPECPREFILL can also be readily combined with other techniques such as speculative decoding, the combination of which could result in the first unified small-model-assisted inference pipeline. With extensive evaluations, we believe that SPECPREFILL will be one of the practical answers to large-scale LLM inference systems.

## Impact Statement

This paper presents work whose goal is to accelerate large language model inference procedures. There are many potential societal consequences of our work, none which we feel must be specifically highlighted here.

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

# A. Standard Short Task Performance of SPECPREFILL

In Figure 6, we report SPECPREFILL when applied to `Llama-3.1-70B-Instruct` on standard short tasks as discussed in Sec 4.6. It is worth noting that for shorter tasks, the queries are more likely to become information dense, rendering SPECPREFILL less effective especially for certain tasks.

# B. Comparing SPECPREFILL with RAG Based Systems

In this section, we first detail the algorithm behind two of our RAG baselines and present results comparing SPECPREFILL against them. Both variants split the context of the prompt into sentences using *nltk* library (Loper & Bird, 2002). After splitting the context, all sentences are encoded using pretrained sentence embedding models (Reimers & Gurevych, 2019). A specially chosen query is used to select relevant sentences based on similarity scores without exceeding the predefined budget. Finally, the new context is re-assembled and fed to the main model. We highlight the key differences in various steps of the pipeline in the following table 3:

| Model Name | Embedding Model | RAG Query | Reassemble Method |
|---|---|---|---|
| RAG-LLAMA LS | `gtr-t5-large` | Last sentence in full prompt | Original order |
| RAG-LLAMA EQ | `all-mpnet-base-v2` | Provided by the dataset | Ordered by relevance |

*Table 3.* **RAG Baseline Specification:** We implemented two RAG baselines with different trade-offs to compare SPECPREFILL.

In Figure 7, we compare `Llama-3.1-70B-Instruct` with RAG-LLAMA-70B LS and RAG-LLAMA-70B EQ. Since both RAG variants are based on sentence chunkation, and hence we calculate the final real token keep percentage for visualization and a fair comparison.

It is worth noting that both two RAG variants can in principle fall short under given tasks due to the fact that their strategy for selecting the query for retrieval is not flexible enough. For example, RAG-LLAMA-70B LS will become less effective when the real query is not placed at the end of the prompt, and RAG-LLAMA-70B EQ not only assumes that the real query is separated from the context and given to the model but also needs special catering when it is not obvious how to design the query for certain tasks (e.g. summarization). Therefore, we consider RAG systems and SPECPREFILL to be useful for different cases with varying degrees of requirements for efficiency, cost, performance, and generality.

# C. Experiment Details

There are some details for our experiments that we wish to give some accounts for:

1. For standard short task evaluation in Sec 4.6, we use LM-EVAL-HARNESS (Gao et al., 2024a) and EVAL-PLUS (Liu et al., 2023). For several tasks in LM-EVAL-HARNESS, we include task configuration files for `Llama-3.1` based on its templates in our code base, which should be placed in the right place for reproducing experimental results.

2. In the QPS experiment in Sec 4.7, we add an extra 5 seconds to the timeout in order to avoid potential system instability. The final results are reported with the original timeout. We set the number of samples for each category based on the maximal QPS we want to evaluate, which makes sure we have a constant QPS during the duration of querying.

3. When running all experiments in vLLM (0.6.3.post1), we set enforce_eager=True and enable_chunked_prefill=False to avoid any unexpected behaviors.

# D. System Specification

In Table 4, we list the detailed specification of the system on which we test the efficiency of models.

# E. Overhead Analysis

SPECPREFILL uses a smaller model as a speculator to help accelerate the larger model. Although proven to be effective, there is no free lunch. In this section, we analyze and quantify the overhead incurred by the speculator so that practitioners are more informed when they choose to use it.

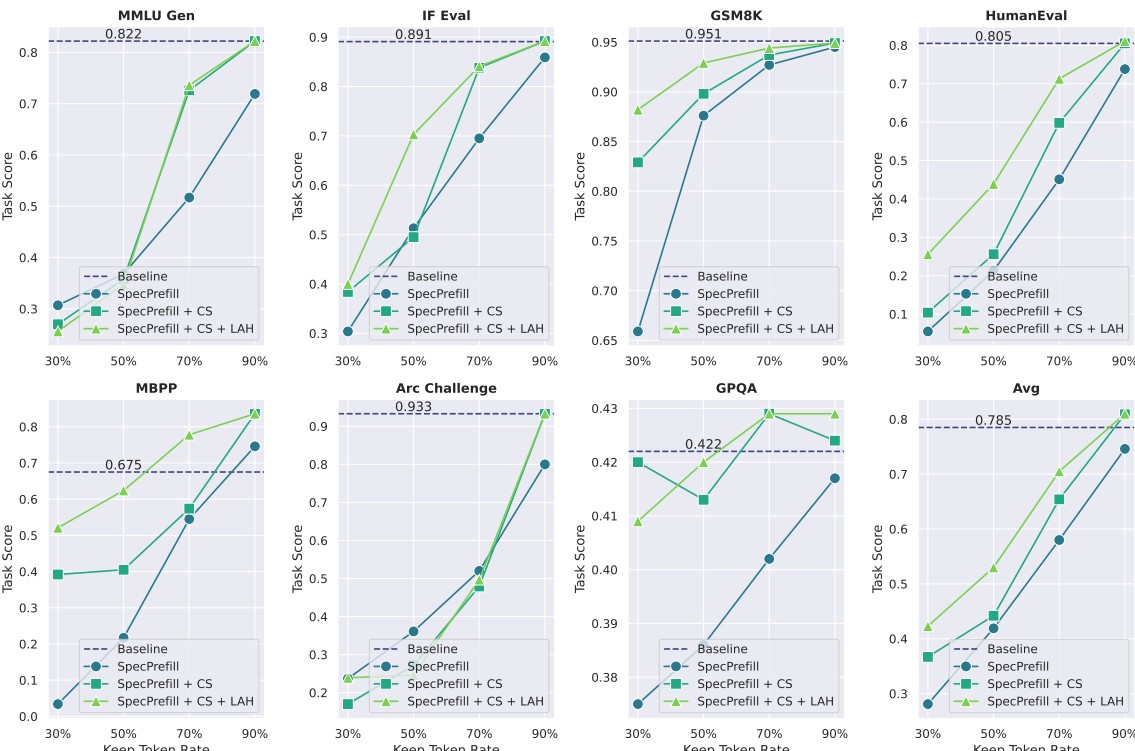

*Figure 6.* **Standard Short Tasks on Llama 70B:** We include results on popular regular context tasks ranging from common knowledge, math, reasoning, and coding ability. Unlike prior works on token eviction and prompt compression, we wish to give a comprehensive evaluation on domains where SPECPREFILL becomes less effective due to the fact that short and knowledge rich prompts are less compressible.

We start by calculating the FLOPS of a transformer model based on a standard implementation, and then compute the theoretical overhead when using a specific speculator for a specific main model with the official Llama model configuration. Since the prefill phase is mostly compute-bound, it is fair to use the FLOPS ratio as a decent approximation of latency improvement. We introduce several parameters for an Llama-like transformer architecture and we calculate the FLOPS of each module separately using these parameters. We ignore less essential computations (e.g. normalization, RoPE, etc) and the formula are shown in Table 5.

In Figure 9, we calculate the theoretical upper-bound of the TTFT speedup for SPECPREFILL with sequence length being $32K$ and batch size equal to 16. As we can see from the theoretical analysis, the real speedup we obtain from the implementation is very close to it, as shown in Figure 3, which suggests that our implementation is highly efficient (i.e. measured $7.66\times$ compared to analyzed $7.72\times$). On each bar, we annotate the overhead of the speculation process, which we define as:

$$overhead(\alpha) := \frac{FLOPS(spec)}{FLOPS(spec) + \alpha * FLOPS(base)}, \forall \alpha \in (0, 1]$$

Within our expectation, the overhead is higher when we have a lower keep rate and lower when we have a higher keep rate. Table E reports the theoretical relative FLOPS between the speculator and the base model.

## F. LongBench Task Length Distribution

We visualize the LongBench suite's average length for each task, which, when coupled with the token keep rate, provides a more clear picture of the model's efficiency gain.

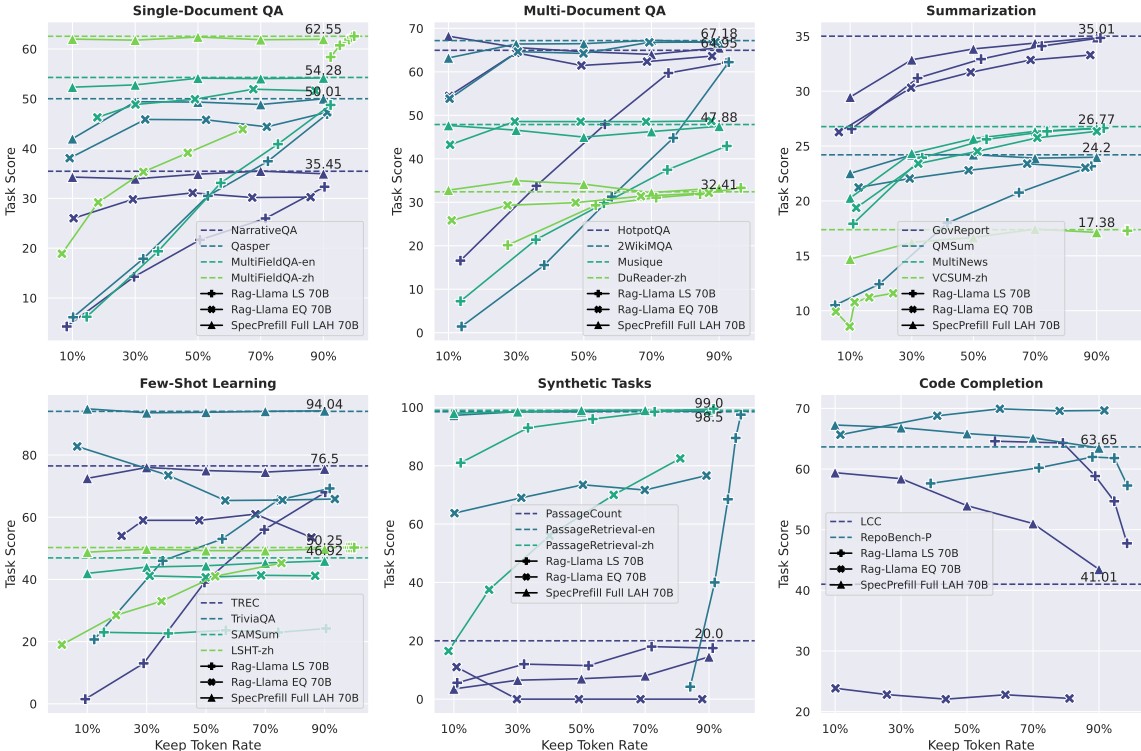

*Figure 7.* **LongBench Baseline Comparison:** We compare SPECPREFILL against the baseline and RAG-LLAMA in this separate figure for clarity. RAG-LLAMA could be effective on certain tasks but for the majority of the tasks, but SPECPREFILL does not require any prior knowledge about the prompt while still being accurate with a finer-control.

| System Hardware/Software Name | Value |
|---|---|
| CUDA Version | 12.7 |
| vLLM Version | 0.6.3.post1 |
| GPU Type | $8 \times$ NVIDIA H200 |
| Total GPU TFLOPS | 428.2 |
| Total RAM | 1123.2 GB |
| Per GPU Memory Bandwidth | 4052.8 GB/s |
| Per GPU NVLink Bandwidth | 478.1 GB/s |
| Per GPU PCIe Bandwidth | 52.8 GB/s |
| Per GPU PCIe Lanes | $16 \times$ PCIe 5.0 |
| Disk Bandwidth | 4730 MB/s |
| Internet Upload Speed | 605.5 Mbps |
| Internet Download Speed | 733.4 Mbps |

*Table 4.* **System Specification for Efficiency Benchmarking:** Efficiency scores can vary when benchmarked on different platforms, and therefore, we list the detailed specification of the system we're using for better reproducibility and understanding.

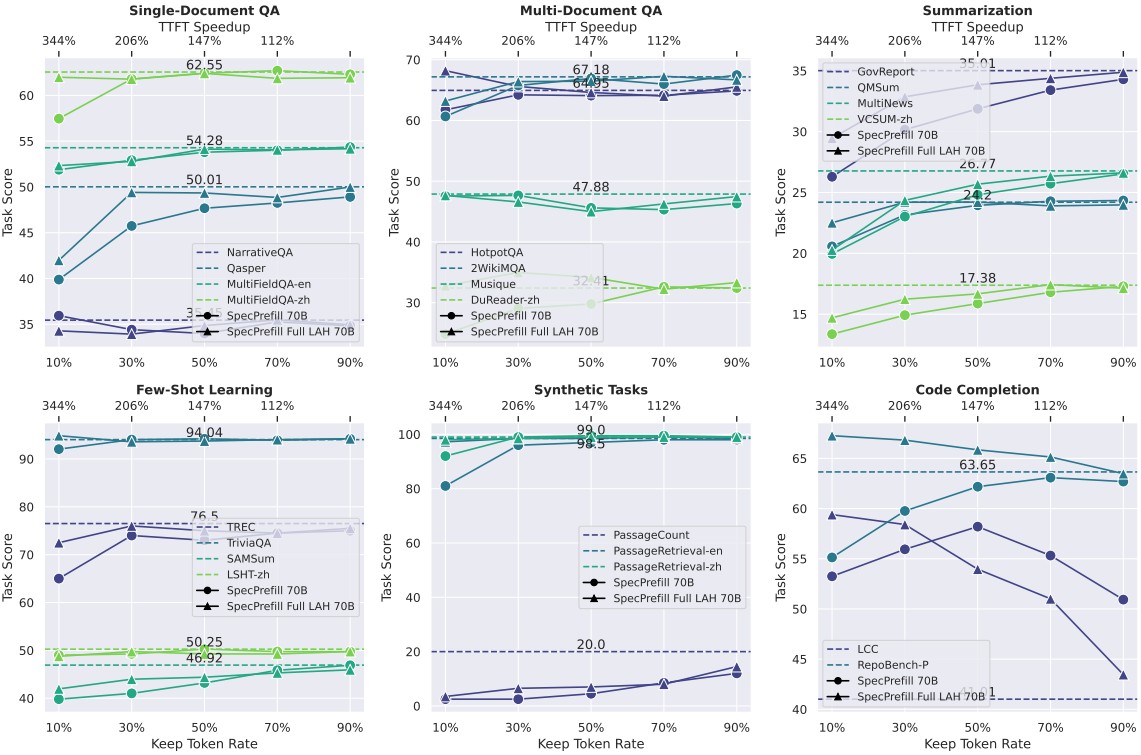

*Figure 8.* **LongBench Results on 70B Model:** We supplement the evaluation of `Llama-3.1-70B-Instruct`, under the same setup as in Figure 2 and 7.

| Parameters | Value |
|---|---|
| Number of layers | $L$ |
| Hidden Size | $D$ |
| Intermediate Size | $I$ |
| Number of Query Heads | $H$ |
| Number of KV Heads | $H'$ |
| Vocabulary Size | $V$ |
| Sequence Length | $S$ |
| Batch Size | $B$ |
| MLP FLOPS | $3BSDI$ |
| QKVO Projections FLOPS | $BSD^2(2 + 2H'/H)$ |
| Self-Attention FLOPS | $2BS^2D$ |
| LM-Head FLOPS | $BSDV$ |
| Total FLOPS | $LBSD(3I + D(2 + 2H'/H) + 2S) + BSDV$ |

*Table 5.* **FLOPS Estimation of Llama Models:** We estimate the FLOPS (MACS) for Llama model with a given configuration. We ignored lower order terms of computations such as vector addition, RMSNorm, RoPE, and treat all matrix operation as FMA for simplicity.

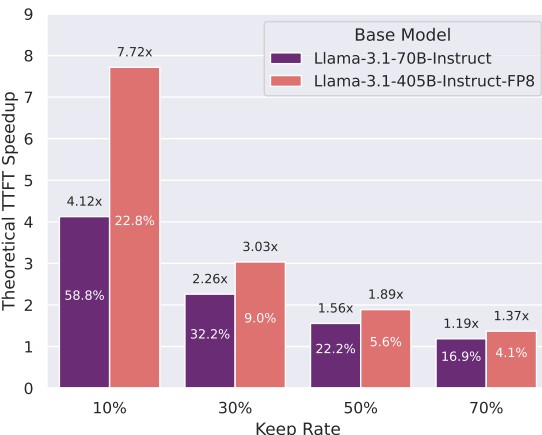

*Figure 9.* **Theoretical TTFT Speedup of SPECPREFILL:** We show the theoretical upper-bound of the TTFT speedup that SPECPREFILL can achieve assuming no other system overhead. The percentage of each bar is the percentage of FLOPS spent on speculation w.r.t. the total FLOPS. Comparing it with results in Figure 3, our real measurement has very little gap to the theoretical maximum, which suggests the efficiency of the implementation.

| **Base Model Size** | $FLOPS_{spec}/FLOPS_{base}$ |
|:---:|:---:|
| 70B | 14.24% |
| 405B | 2.96% |

*Table 6.* **Relative FLOPS of SPECPREFILL:** We calculate the theoretical FLOP ratio between the speculator model of size 8B and the base model.

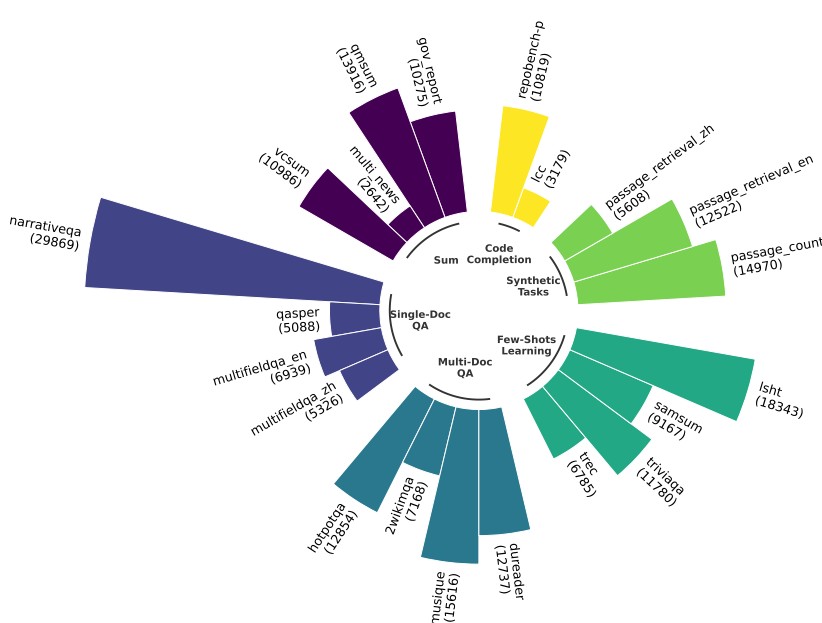

*Figure 10.* **LongBench Prompt Token Lengths:** We visualize the average token lengths of prompts for each task spanning the five major categories in LongBench.

