# OpenReview forum: "Speculative Prefill: Turbocharging TTFT with Lightweight and Training-Free Token Importance Estimation"
_ICML.cc/2025/Conference — ICML 2025 poster_

### Official Review · Reviewer_JHsj · 2025-03-06

**Overall Recommendation:** 2

**Summary:**

SpecPrefill proposes a prompt token pruning technique using speculation by a small model before forwarding the large model. This method is especially beneficial when the token length is medium (latency of attention and MLP are similar, so sparse attention methods do not do a good job) because it reduces the number of tokens.

**Claims And Evidence:**

They have some evidence, but I cannot find any comparison between baseline. They have only vanilla attention, while they still can compare their method with other pruning methods (such as sparse attention, MInference: https://github.com/microsoft/MInference, FlexPrefill: https://arxiv.org/abs/2502.20766)

**Essential References Not Discussed:**

I think they should compare it with the sparse attention method, even if they claim that MLP is a bottleneck in the middle and a little bit longer in context. I cannot fully agree that MLP is a significant bottleneck, so speeding up attention is not important, as declared in lines 108 and 121. I partially agree that MLP is a bottleneck in a somewhat short context (less than 8K tokens). However, these days, many applications require longer context to perform well (e.g., RAG, Image (costs 1-4K tokens per image), and Video (costs 100-1K tokens per frame) model). Therefore, I suggest the  following works (that can speed up prefill) to compare:
- MInference: https://github.com/microsoft/MInference
- SnapKV: https://github.com/FasterDecoding/SnapKV
- HiP Attention: https://github.com/DeepAuto-AI/hip-attention
- SampleAttention: https://arxiv.org/html/2406.15486
- DuoAttention: https://github.com/mit-han-lab/duo-attention
- FlexPrefill: https://openreview.net/forum?id=OfjIlbelrT

**Experimental Designs Or Analyses:**

I think an ablation study on N should be in the paper, which is used in algorithm 1 line 3.

**Methods And Evaluation Criteria:**

I think the method looks correct. However, the evaluation is weak due to a lack of baselines.

**Other Comments Or Suggestions:**

Please refer to other sections.

**Other Strengths And Weaknesses:**

I think the tested context lengths are too short compared to other modern LLM (Qwen 2.5=1M tokens, Llama 3.1=128K tokens ...) I think we should try at least up to 128K tokens, which is the context length of Llama3.1. The RULER and InfintieBench (https://github.com/OpenBMB/InfiniteBench) benchmark will be a great choice to test the effectiveness

**Questions For Authors:**

Please refer to other sections.

**Relation To Broader Scientific Literature:**

The key contribution is about speeding up the QPS of the LLM serving framework, which is widely used in various NLP research. I think this kind of topic is pretty important, and this method is tackling that issue.

**Theoretical Claims:**

Yes.

---

> ### Author Rebuttal · Authors · 2025-04-01
>
> ### Summary
> We sincerely thank the reviewer for the effort in providing constructive comments. We addressed all the feedback from the reviewer, and believe that with these changes, the results in our paper become significantly stronger (added experiment result numbers in reviewer USDK’s rebuttal).
>
> ### Per-feedback detailed response
> > I cannot find any comparison between baseline.
>
> **In the original submission, we included two RAG baselines where we put the conclusion in the experiment section 4.4 and detailed results in the Appendix B. We now move all results in T1 (reviewer USDK's comment) to the main body, along with two more baselines as suggested by the reviewer**:
>
> 1. We evaluate LLMLingua 2 using LongBench. Similar to RAG, LLMLingua does not support query-aware compression, and therefore, we separate the context from the query and send it to the compression model. **In T1 (reviewer USDK's comment), SpecPrefill outperforms LLMLingua and RAG Llama**.
>
> 2. We benchmark MInference's quality on LongBench and its inference latency in vLLM for a more comprehensive view. In T1 (reviewer USDK's comment), the quality score of MInference can already be matched very closely with SpecPrefill with 10% token keep rate (0.3% loss) and is outperformed when we increase the rate to 30% and higher. **In T2 (reviewer USDK's comment) where we vary the batch size x sequence length products from 128 * 4K to 4 * 128K, we found that SpecPrefill is particularly effective in efficiency compared to MInference with similar or better quality (getting 2.54x - 6.54x relative speedup compared to MInference)**. The main reason why MInference might fall short is due to the overhead in dynamic index building, which gets significant when the batch size is large (this is also observed in paper https://arxiv.org/abs/2502.05431 and confirmed by the author in issue https://github.com/microsoft/MInference/issues/18).
>
> > I think an ablation study on N should be in the paper.
>
> We agree that ablation on N will provide more insights and are running a full ablation on N for the camera ready version. **For our current results as in Figure 2 and Figure 5, we showcase the benefits of using LAH with steps = 8 (with LAH as the suffix in model names), which proves its usefulness especially for shorter context tasks**.
>
> > I strongly suggest authors submit the code as a supplementary to find potential practical issues rather than showing it in only papers.
>
> We agree on this and we will upload code whenever OpenReview allows uploading again. We’re confident that our experiments are reproducible and implementations are highly optimized (see Appendix E).
>
> > I cannot fully agree that MLP is a significant bottleneck, so speeding up attention is not important, as declared in lines 108 and 121. I partially agree that MLP is a bottleneck in a somewhat short context (less than 8K tokens).
>
> Sorry for the confusion! We fully agree that attention is a significant bottleneck (in both short-context and especially long context) and SpecPrefill also helps skip the attention part as well, as shown in T2 (reviewer USDK’s comment) when we move on to 128K. **What we hoped to convey was that there are many existing techniques to optimize attention in various regimes but optimizing MLP hasn't attracted too much attention, which is where SpecPrefill stands out**. We revised the wordings to make this point clear.
>
> > I think we should try at least up to 128K tokens... The RULER and InfintieBench benchmark will be a great choice to test the effectiveness.
>
> We understand the reviewer’s concern on longer context performance. **We increase the RULER results in the original paper from 32K to 64K and observe consistent trends (T3 reviewer USDK’s comment)**. We will present the result on 128K for the camera-ready version and believe the same conclusion to hold (it is still currently running due to large models and limited cards).
>
> > Essential References Not Discussed.
>
> We thank the reviewer for providing relevant works:
> 1. MInference, SnapKV, HiP Attention, SampleAttention, DuoAttention, and FlexPrefill are awesome work. **However, they focus primarily on the attention module while keeping the MLP part intact. We include those in our camera-ready version for more context**.
> 2. **For MInference, we added additional experiments to show that SpecPrefill achieves better speedup with better quality**.
> 3. **MInference, SnapKV, and DuoAttention were originally in our draft** and we added more context and discussion in our camera-ready version.
>
> We truly agree with the concerns and feedback from the reviewer and hope our responses can help address some of those. With the added results, we are confident that SpecPrefill can be applied and benefit many real world applications and the contributions are novel and fully reproducible. We’d love to hear from the reviewer on further suggestions if any to make the paper better!

---

> > ### Comment · Reviewer_JHsj · 2025-04-03
> >
> > Thank you for the rebuttal. However, I still have concerns regarding the empirical evaluation.
> >
> > > MInference, SnapKV, and DuoAttention were originally in our draft
> >
> > I could not find any results or evaluations involving MInference, SnapKV, or DuoAttention in the original draft. If I missed them, please clarify where they appear.
> >
> > > For MInference, we added additional experiments to show that SpecPrefill achieves better speedup with better quality.
> >
> > I appreciate the added experiments. However, the overall evaluation still lacks sufficient empirical comparison against relevant baselines (Still lacks the baselines in RULER; just two baselines are provided for LongBench with only a single model family, no detailed latency reports, and reduction analysis). The draft would benefit significantly from a more comprehensive and balanced comparison across a wider range of methods.
> >
> > > (it is still currently running due to large models and limited cards).
> >
> > If resource constraints are a challenge, I wonder why the authors chose to focus primarily on very large models such as 70B and 405B. While these models are impressive, evaluating on smaller models (e.g., 8B or 14B) could help demonstrate the effectiveness of the proposed method more broadly and allow for more complete comparisons under limited compute.
> >
> > > enforce_eager=True in L923
> >
> > This raises a critical concern. Disabling graph compilation can significantly impact decoding latency and overall throughput. Since QPS is a major performance metric in this paper, this decision may lead to exaggerated runtime measurements. The potential impact on reported results should be carefully examined and discussed.
> >
> > Finally, while the additional Table T1 from the USDK evaluation is a helpful step, it is not sufficient to address the broader concern about limited baseline comparisons. I strongly encourage the authors to provide a more thorough empirical study across a range of tasks and methods to better support the claims of the paper.

---

> > > ### Author Response · Authors · 2025-04-05
> > >
> > > Thank you for expressing your concern!
> > >
> > > ### Code with Reproducible Experiments
> > > Here is our [code](https://anonymous.4open.science/r/speculative_prefill-1E26)
> > >
> > > ### Answers to Concerns
> > > > I could not find any results or evaluations involving MInference, SnapKV, or DuoAttention in the original draft.
> > >
> > > They were properly **cited and discussed (L119) as suggested by the initial review**. We added more discussion below and in our final version.
> > >
> > > > enforce_eager=True in L923
> > >
> > > 1. enforce_eager=True will not change anything about prefill. Second, **vLLM has a buggy implementation for this for many versions**:
> > > - So many [Issue 1](https://github.com/vllm-project/vllm/issues/10559), [Issue 2](https://github.com/vllm-project/vllm/issues/13536),  [Issue 3](https://github.com/vllm-project/vllm/issues/9670), [Issue 4](https://github.com/vllm-project/vllm/issues/8494), [Issue 5](https://github.com/vllm-project/vllm/issues/2229) etc about: wrong output related to eager mode.
> > > - [Issue](https://github.com/vllm-project/vllm/issues/3874) about memory.
> > > - Minference uses enforce_eager=True as well: [see](https://github.com/microsoft/MInference/blob/9ee9d4bbc725a5ab8c992a8a6cbce3601c6e8deb/experiments/benchmarks/benchmark_e2e_vllm_tp.py#L65)
> > >
> > > 2. **We explicitly mentioned the reason for doing so in Appendix C.**
> > >
> > > 3. The reason we found this was that we previously tried setting it to False and gave us **wrong** outputs, but the improvement on QPS is consistent because it is mostly bounded by the TTFT as analyzed in section 4.6.1 (since for most tasks the prompts are way longer than the outputs). All other efficiency plots are setting output_len=1, so **the results won’t be different**.
> > >
> > > > If resource constraints are a challenge …
> > >
> > > We believe that trying large models is an advantage rather than disadvantage, showing the scaling power of SpecPrefill under the most demanding applications.
> > >
> > > ### About Our Current Results
> > >
> > > > However, the overall evaluation still lacks sufficient empirical comparison against relevant baselines…
> > >
> > > 1. We evaluated (T1 from USDK’s comment) **four** instead of two baselines:
> > > - Instruct baseline
> > > - Rag baseline (two variants in Appendix B)
> > > - LLMLingua (text compression)
> > > - Minference (sparse attn).
> > >
> > > 2. **We spent the whole section 4.6 + T2 (USDK) + Appendix E for the efficiency report** with real world setup:
> > > - vLLM concurrency + large models.
> > > - Consistent QPS flows + API communication.
> > > - Real queries with scheduling + large batch size.
> > >
> > > 3. Our central claim that **SpecPrefill greatly improves TTFT (and hence maximal QPS) with minimal loss in quality** has been sufficiently shown in LongBench & RULER even just by comparing with baseline alone (>= 64K below) (we did not overclaim betterness than non-comparable methods, see in next section):
> > >
> > > |  |  | Retrieval | Multi-hop | QA | Aggregation | Avg w/o Agg |
> > > |---|---|---|---|---|---|---|
> > > | baseline 70B | 64k | 98.5 | 99.9 | 65.1 | 65.6 | 87.9 |
> > > |  | 128k | 76.5 | 56.1 | 48.2 | 41.3 | 60.3 |
> > > | SpecPrefill 10% | 64k | 99.5 | 99.8 | 71.9 | 54.9 | 90.4 |
> > > |  | 128k | 85.8 | 55.6 | 55.3 | 48.3 | 65.6 |
> > >
> > > ### About Baselines
> > >
> > > To begin with, we hope to first convince the reviewer why we included some and didn’t include other baselines.
> > >
> > > Any efficiency arguments about inference should be made based on highly optimized impl for a fair comparison:
> > > - We chose MInference because it has vLLM support with 70B.
> > > - We chose RAG and LLMLingua because they preprocess the inputs and can feed directly to vLLM servers.
> > > - SnapKV **doesn’t improve prefill**, and is written in [HF only](https://github.com/FasterDecoding/SnapKV/blob/main/experiments/LongBench/pred_snap.py).
> > > - DuoAttention falls to the same category of MInference we tested, is written in [HF](https://github.com/mit-han-lab/duo-attention/blob/main/scripts/efficiency.sh) only, and evaluated with [bs = 1](https://github.com/mit-han-lab/duo-attention/blob/fe93c314ae87306ef6629dc16713250b4718ffe7/eval/efficiency/benchmark_static.py#L46)
> > >
> > > Running any model of the same size in vLLM will surely be faster than in HF by a magnitude. Implementing a fast and scalable framework itself should be considered as a part of our core contribution rather than disadvantage. And **we are not responsible for providing optimized implementation for other methods**.
> > >
> > > **Nonetheless, we give a super detailed analysis to compare DuoAttn and SnapKV [here](https://anonymous.4open.science/r/speculative_prefill-1E26/COMPARE.md)**.
> > >
> > > All results support our main claim that SpecPrefill makes QPS + Prefill faster with minimal quality loss and scales to 405B. And we’re confident that current results (including newly added baselines) positively support the effectiveness of SpecPrefill with the most fair and realistic setting.
> > >
> > > Could you please double-check our results and re-evaluate the comprehensiveness and soundness of experiments? We are happy to hear more feedback.

---

### Official Review · Reviewer_USDK · 2025-03-08

**Overall Recommendation:** 3

**Summary:**

This paper introduces Speculative Prefill, an innovative training-free framework that elegantly addresses efficiency challenges in language model inference. By leveraging a lightweight model to speculate on important tokens based on context, the approach impressively enhances time-to-first-token performance. The results are quite remarkable, achieving up to 7x higher queries per second and a 7.66x improvement in TTFT.

**Claims And Evidence:**

The paper presents its claims with well-supported evidence throughout, providing a convincing foundation for the proposed approach.

**Essential References Not Discussed:**

I haven't identified any significant omissions in the paper's references.

**Experimental Designs Or Analyses:**

The experimental design demonstrates careful consideration across multiple dimensions. The comparison of different token keep percentages on LongBench and Ruler offers valuable insights, while the efficiency evaluations provide a holistic view of the method's practical benefits. Overall, the experimental approach nicely balances breadth and depth.

**Methods And Evaluation Criteria:**

The method strikes an excellent balance between simplicity and effectiveness, offering an elegant solution to the prefill optimization problem.
The evaluation framework thoughtfully considers both task performance scores and TTFT metrics, which provides a comprehensive view of the inevitable trade-offs between accuracy and efficiency that such approaches entail.

**Other Comments Or Suggestions:**

No additional comments.

**Other Strengths And Weaknesses:**

Strength:
1. The speculative prefill concept is particularly compelling, offering an insightful approach to identifying important tokens using a smaller model's predictions as input for the larger model.
2. The paper presents its ideas with clarity and thoughtful organization.
3. The experimental section offers comprehensive and nuanced analysis of the method's performance.

Weakness:
1. While the approach is quite innovative, it would be interesting to see more exploration of how it compares with alternative attention sparsity methods such as minference, especially considering the different optimization strategies they employ.

**Questions For Authors:**

1. I'm curious about how this approach might compare with attention sparsity methods like minference. Both approaches optimize different aspects of the inference process, and while this method also addresses MLP computation, I wonder if you could share any insights or perhaps preliminary comparisons on the relative advantages of each approach? This could help readers better understand the positioning of your contribution within the broader landscape of inference optimization techniques.

**Relation To Broader Scientific Literature:**

Not addressed in this review.

**Theoretical Claims:**

The theoretical underpinnings of the work appear sound and well-reasoned, with clear connections between the conceptual framework and practical implementation.

---

> ### Author Rebuttal · Authors · 2025-04-01
>
> ### Summary
> We thank the reviewer for the constructive and insightful feedback! We addressed all feedback and believe that incorporating all comments makes this paper significantly stronger:
>
> ### Per-feedback detailed response
> > It would be interesting to see more exploration of how it compares with alternative attention sparsity methods such as Minference…
>
> As the reviewer has mentioned, SpecPrefill not only reduces the computation on the attention but also on the MLP while sparse attention primarily focuses on the attention module alone. In order to provide a more comprehensive comparison between the two routes, we conducted experiments comparing ours against MInference as requested by the reviewer.
>
> We adopt the pre-searched optimal pattern from MInference’s official repo for the 70B model and benchmark’s its quality on LongBench in T1 (see below). As we can see **SpecPrefill can achieve 99.7% quality of MInference while keeping only 10% tokens and surpasses it when using 30% or more tokens**. Because MInference is an attention based sparsity optimization, we benchmark its latency for a fair comparison in terms of acceleration and quality. In T2 (see below) where we vary the batch size x sequence length products from 128 * 4K to 4 * 128K, we found that **SpecPrefill is particularly effective in efficiency compared to MInference with similar or better quality (getting 2.54x - 6.54x relative speedup compared to MInference)**. The main reason why MInference might fall short is due to the overhead in dynamic index building, which gets significant when the batch size is large. We do want to emphasize that Minference is an excellent work and its advantage starts to shine when the sequence becomes ultra long, at least 128K+ (or the ratio between sequence length and batch size gets large enough). Another trade-offs we want to mention between our methods and sparse attention approaches are the decision of whether we want to compute / store all the KV caches. Besides, MLP generally dominates the prefill part for large batch size + decent context lengths (at least within our tested 128K). This becomes essentially important based on the application scenario (e.g whether the prefill/decode is disaggregated and hence requires KV transmission). Overall, we believe that since both methods require no-training and can be switched on and off on the fly, they are complementary to each other and can hence meet a wider range of requirements when combined together.
>
> With the added results, we are confident that SpecPrefill can be applied and benefit many real world applications and the contributions are novel and fully reproducible. We’d love to hear from the reviewer on further suggestions if any to make the paper better!
>
> ### Newly added experiment results:
>
> **LongBench comparison with baselines (T1):**
> | Model | Compression Rate | Single-Doc QA | Multi-Doc QA | Sum | Few-shot Learning | Code | Synthetic | Avg |
> |---|---|---|---|---|---|---|---|---|
> | baseline 70B | N/A | 50.57 | 53.11 | 25.84 | 66.93 | 52.33 | 72.50 | 53.55 |
> | RAG | 10.90% | 32.32 | 41.17 | 18.86 | 45.40 | 44.76 | 30.42 | 35.49 |
> |  | 28.60% | 38.43 | 47.41 | 21.42 | 50.53 | 45.80 | 35.50 | 39.85 |
> |  | 46.80% | 40.53 | 46.64 | 22.45 | 49.52 | 46.00 | 43.15 | 41.38 |
> |  | 65.10% | 41.40 | 47.43 | 23.30 | 52.21 | 46.19 | 47.22 | 42.96 |
> |  | 83.40% | 43.25 | 48.16 | 23.56 | 51.44 | 45.92 | 53.04 | 44.23 |
> | LLMLingua | ~10% | 26.50 | 32.94 | 20.95 | 37.40 | 45.00 | 16.33 | 29.85 |
> |  | ~30% | 38.83 | 44.02 | 23.37 | 42.23 | 47.27 | 37.00 | 38.79 |
> |  | ~50% | 43.64 | 50.67 | 24.77 | 50.96 | 49.05 | 60.33 | 46.57 |
> |  | ~70% | 45.90 | 52.88 | 25.44 | 59.77 | 51.48 | 68.50 | 50.66 |
> |  | ~90% | 45.94 | 53.91 | 25.87 | 60.46 | 54.06 | 72.00 | 52.04 |
> | Minference | N/A | 50.46 | 53.23 | 25.83 | 66.36 | 52.48 | 69.00 | 52.89 |
> | SpecPrefill | 10% | 47.64 | 52.96 | 21.74 | 64.52 | 63.33 | 66.25 | 52.74 |
> |  | 30% | 49.47 | 53.39 | 24.41 | 65.83 | 62.62 | 67.83 | 53.92 |
> |  | 50% | 50.18 | 52.56 | 25.10 | 65.60 | 59.91 | 68.17 | 53.59 |
> |  | 70% | 50.06 | 52.44 | 25.51 | 65.77 | 58.08 | 68.67 | 53.42 |
> |  | 90% | 50.26 | 53.25 | 25.65 | 66.35 | 53.47 | 70.67 | 53.27 |
>
> **Latency comparison (T2):**
> | | 128 * 4k | 64 * 8k | 32 * 16k | 16 * 32k | 8 * 64k | 4 * 128k |
> |-----------------|----------|---------|----------|----------|---------|----------|
> | 70B Instruct | 22.6 | 23.7 | 25.9 | 30.1 | 38.7 | 56.0 |
> | Minference | 46.6 | 45.3 | 42.5 | 40.8 | 38.3 | 34.6 |
> | SpecPrefill 10% | 7.1 | 7.0 | 7.2 | 8.0 | 9.8 | 13.6 |
> | SpecPrefill 30% | 11.8 | 11.7 | 12.1 | 13.2 | 15.7 | 20.9 |
> | SpecPrefill 50% | 16.2 | 16.4 | 17.1 | 19.0 | 22.7 | 30.7 |
>
> **RULER 64K (T3):**
> |  | Retrieval | Multi-hop | QA | Agg | Avg w/o Agg |
> |---|---|---|---|---|---|
> | baseline 70B | 98.5 | 99.9 | 65.1 | 65.6 | 87.9 |
> | SpecPrefill 10% | 99.5 | 99.8 | 71.9 | 54.9 | 90.4 |

---

> > ### Comment · Reviewer_USDK · 2025-04-02
> >
> > Thank you for your reply. I will keep my score.

---

> > > ### Author Response · Authors · 2025-04-05
> > >
> > > Thank you for recognizing our work. According to Reviewer JHsj's comment, we further added **comparison with DuoAttention and SnapKV with 8B model here: [analysis](https://anonymous.4open.science/r/speculative_prefill-1E26/COMPARE.md)** and **128K RULER** in JHsj's comment.
> > >
> > > We also **uploaded our [code](https://anonymous.4open.science/r/speculative_prefill-1E26/README.md)** in case it can help better assess our work's validity.
> > >
> > > We appreciate your time in reviewing our work!

---

### Official Review · Reviewer_pMkJ · 2025-03-11

**Overall Recommendation:** 3

**Summary:**

This paper propose SpecPrefill, that identifies the important tokens during the pre-fill stage via accumulating the attention scores in a 8-steps look-ahead window, calculated with a small model as the speculator. This approach can achieves a speedup of about 3x and also maintains a comparable performance compared to full attention.

**Claims And Evidence:**

-

**Essential References Not Discussed:**

You should cite:
Li, Y., Dong, B., Lin, C., & Guerin, F. (2023). Compressing Context to Enhance Inference Efficiency of Large Language Models. Conference on Empirical Methods in Natural Language Processing.
for prompt compression in addition to llmlingua series of model.

You should test the proposed method in SCBench to see how well the method perform under multi-turn scenarios:
Li, Y., Jiang, H., Wu, Q., Luo, X., Ahn, S., Zhang, C., Abdi, A.H., Li, D., Gao, J., Yang, Y., & Qiu, L. (2024). SCBench: A KV Cache-Centric Analysis of Long-Context Methods. ArXiv, abs/2412.10319.

It may also help to show how well the method works on more challenging long-context tasks such as gsm-infini.
Zhou, Yang et al. “GSM-Infinite: How Do Your LLMs Behave over Infinitely Increasing Context Length and Reasoning Complexity?” (2025).

**Experimental Designs Or Analyses:**

This is method aims to optimize the pre-filling stage unlike the existing sparse attention, instead it uses a speculator to find the critical tokens and only involve these in the actual attention computation.

The strength of the paper:
1. good system design. the proposed method reuse the speculative decoding framework, which extend the speculative decoding to the per-fill stage, so the large model not only benefit from the decoding proposal, but also the sparse pre-fill. This also makes this method to be easily adopted in systems with spec-decoing supported.

2. good performance on most of tasks. spec-prefill achieves comparable performance against full attention on long-bench and ruler, with 50% tokens perserved. it achieves good speedup (2x speedup when 50% tokens perserved).

3. a novel attempts for pre-fill acceleration that may inspire future works.

The limitation of the paper:
1. not a single baseline was included in the experiment. This proposed method uses a draft/small model to evaluate the importance of the tokens from the prompt, is pretty much similar to the idea of prompt comparession and kv cache compression. Prompt compression, in particular, also aims to accelerate the pre-filling acceleration, which should definitely be compared against spec-prefill. In fact, spec-prefill is simply a prompt compression technique with a 8-step look-ahead window. In addition, dynamic sparse attention that build sparse index online for faster pre-fill is also very relevant to spec-prefill, which is a must to compare with. the missing of these baselines is a big limitation of this paper.

2. this token-level sparse attention is previously studied to be less robust in real production environment, especially in multi-round scenarios. SCBench demonstrates that the critical tokens of a prompt can be dramatically different for different queries. In other words, the look-ahead window in spec-prefill can be sub-optimal when the conversation goes on and move to the next query. The benchmarks used in the experiments fail to prove that the proposed method can tackle such a limitation.

**Methods And Evaluation Criteria:**

-

**Other Comments Or Suggestions:**

-

**Other Strengths And Weaknesses:**

-

**Questions For Authors:**

1. try compared spec-prefill against prompt compression methods, and dynamic sparse attention approaches.
2. try test spec-prefill on multi-turn long context benchmarks such as SCBench.

**Relation To Broader Scientific Literature:**

-

**Theoretical Claims:**

-

---

> ### Author Rebuttal · Authors · 2025-04-01
>
> ### Summary
> We sincerely thank the reviewer for the time and constructive feedback! We addressed all feedback and believe that it makes this paper significantly stronger (**added experiment result numbers in reviewer USDK’s rebuttal**). Specifically, we addressed the feedback by:
>
> 1. We moved the original main results for our RAG-baselines to the main body and created a table to compare all methods including two newly added baselines: LLMLingua and MInference. SpecPrefill achieves higher quality and better efficiency than all of them.
> 2. We agree with the application of multi-turn situations and believe that SpecPrefill will be able to solve it with the full knowledge of the speculator. We will add results in the camera-ready version (running the experiments).
> 3. We added more context and discussions for the essential references in our camera-ready version.
>
> ### Per-feedback detailed response:
>
> > Not a single baseline was included in the experiment… Try comparing spec-prefill against prompt compression methods, and dynamic sparse attention approaches.
>
> In the previous draft, we compared two RAG-based methods in the Appendix. We should have done a better job in making them more visible -- to do this, we will add a summary of baseline comparison in T1 (reviewer USDK's comment) in the main paper body.
>
> In addition, we follow the advice from the reviewer by adding LLMLingua (text-level prompt compression, https://arxiv.org/abs/2310.05736) and MInference (sparse attention, https://arxiv.org/abs/2407.02490) to complement our original RAG baseline.
>
> We evaluate all three baselines on LongBench, and the result is shown in T1 (reviewer USDK's comment). **SpecPrefill outperforms RAG and LLMLingua by margins when keeping the same tokens from the prompt** (i.e. same tokens ~= similar speedup). Comparing MInference requires a more careful design due to the fact that it is a sparse attention optimization that does not change the computation of the MLP and needs to compute all KVs. Therefore, we benchmark MInference’s prefill latency in T2 (reviewer USDK's comment) where we vary the batch size x sequence length products from 128 * 4K to 4 * 128K. We observe that MInference has substantial overhead when the batch size is large and only gets amortized when the sequence length gets ultra long (at least > 128K with small batch size < 4, this is also observed in paper https://arxiv.org/abs/2502.05431 and confirmed by the author in issue https://github.com/microsoft/MInference/issues/18). **Overall, SpecPrefill can achieve 2.54x to 6.54x relative speedup compared to MInference with 99.7% quality and surpasses MInference’s quality when keep rate is at least 30%**.
>
> > Try test spec-prefill on multi-turn long context benchmarks such as SCBench.
>
> We fully agree that multi-turn tasks are important applications to test SpecPrefill. **Unlike many prompt compression methods, SpecPrefill maintains the full KV for the speculator, which makes it possible to revive dropped tokens during the first round of conversation. This allows us to re-estimate token importance when a new turn comes in and let the base model fill-in the missing KVs as needed**. We are currently running evaluations and will include results for the camera-ready version (due to new implementation and evaluation time, we can’t finish it during the rebuttal window). In addition, here’s our detailed description of what we’re running:
>
> We outline the high-level algorithm here:
> 1. Check if it is the first turn. If so we just call our standard SpecPrefill and return the results. If not, move to step 2.
> 2. Let the speculator first process the new N-token context and estimate important tokens just like in the first prefill.
> 3. Identify missing KVs based on slot-mapping and position information.
> 4. For positions missing KVs, we simply keep those ids.
> 5. For the base model, we will recompute the KVs if some ids are not in the cache. Return the new results.
>
> The total number of KVs that will be computed and stored will not exceed the context length, which means there is no waste of “recomputation”.
>
> Nonetheless, we’d like to mention that our initial starting goal for SpecPrefill is to tackle large-batch offline (no further interaction) prefill acceleration that targets a large traffic of requests such as bulk document summarization, QA, etc.
>
> > Essential reference not mentioned.
>
> All of these are awesome add-ons and have been included in appropriate places for reference.
>
> We are confident that SpecPrefill can be applied and benefit many real world applications and the contributions are novel and fully reproducible. We’d love to hear from the reviewer on further suggestions if any to make the paper better!

---

### Decision · Program_Chairs · 2025-05-01

**Decision:**

Accept (poster)

**Comment:**

I lean toward accepting this paper. While concerns about baseline comparisons were valid, the authors thoroughly addressed these in their rebuttal by adding LLMLingua and MInference comparisons, extending evaluations to longer contexts, and providing their code. The paper tackles the important TTFT problem with a novel approach that achieves substantial speedups while maintaining quality. The method's effectiveness across model scales and its focus on optimizing both attention and MLP computation represent valuable contributions to efficient LLM inference.